# END-TO-END NEURAL NETWORK COMPRESSION VIA $\frac{\ell_1}{\ell_2}$ REGULARIZED LATENCY SURROGATES

## ABSTRACT

Neural network (NN) compression via techniques such as pruning, quantization requires setting compression hyperparameters (*e.g.,* number of channels to be pruned, bitwidths for quantization) for each layer either manually or via neural architecture search (NAS) which can be computationally expensive. We address this problem by providing an end-to-end technique that optimizes for model's Floating Point Operations (FLOPs) via a novel $\frac{\ell_1}{\ell_2}$ latency surrogate. Our algorithm is versatile and can be used with many popular compression methods including pruning, low-rank factorization, and quantization. Crucially, it is fast and runs in almost the same amount of time as a *single model training run*; which is a significant training speed-up over standard NAS methods. For BERT compression on GLUE fine-tuning tasks, we achieve $50\%$ reduction in FLOPs with only $1\%$ drop in performance. For compressing MobileNetV3 on ImageNet-1K, we achieve $15\%$ reduction in FLOPs *without drop in accuracy*, while still requiring $3\times$ less training compute than SOTA NAS techniques. Finally, for transfer learning on smaller datasets, our technique identifies $1.2\times$-$1.4\times$ cheaper architectures than standard MobileNetV3, EfficientNet suite of architectures at almost the same training cost and accuracy.

## 1 INTRODUCTION

Large-scale neural networks consistently provide state-of-the-art performance on complex learning tasks (He et al., 2016; Tan & Le, 2019; Kaplan et al., 2020). But they place heavy burden on compute resources such as battery, memory or processor making them hard to deploy on edge devices such as phones, cameras and wearables. Several recent works have designed techniques to compress ML models and make them efficient for inference. However, as detailed below, many of these techniques are hard to use in practice, and often achieve sub-optimal accuracy *vs* inference time trade-offs.

**Hyperparameter search for compression.** Existing works typically rely on one of the following building blocks to design efficient models: unstructured weights sparsity (Han et al., 2015; Kusupati et al., 2020; Tiwari et al., 2021), pruning entire neurons or low-rank factorization (Wang et al., 2019; Hsu et al., 2021), quantization (Nagel et al., 2021), distillation (Bucila et al., 2006; Hinton et al., 2015). Figuring out an optimal way to combine these building blocks (or to figure out hyper-parameters such as amount of sparsity associated with each block) while satisfying a global FLOPs/latency/resource constraint is difficult and involves a combinatorial search. This problem is further exacerbated when multiple building blocks are used for model compression (*e.g.,* simultaneous low rank factorization, sparsity/pruning of weights).

Over the past few years, there has been a large body of work that addresses the problem of finding hyperparameters for model compression. Existing literature in this space can be broadly classified into two categories depending on the style of optimization techniques employed: blackbox, and whitebox techniques.

**Blackbox Compression Techniques.** Several works in this category formulate model compression as a black-box Neural Architecture Search (NAS) problem and rely on state-of-the-art NAS techniques to search for efficient models (Zoph & Le, 2016; Kandasamy et al., 2018; Yang et al., 2018). These techniques directly take the FLOPs/latency into account and have the potential to identify the optimal per-layer budget allocation for a wide variety of efficient blocks/compression mechanisms. However,

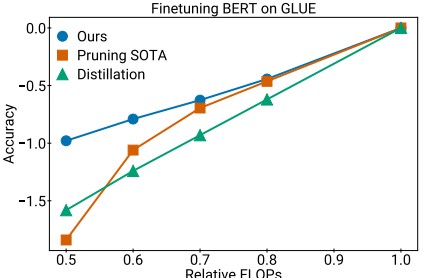 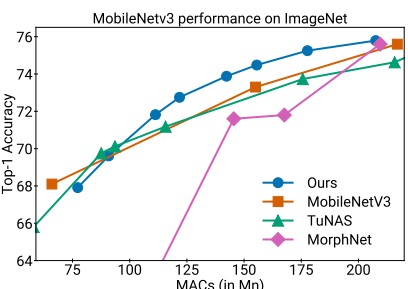

Figure 1: Left plot compares various techniques for BERT compression on GLUE tasks (averaged across tasks). $x$-axis is the relative number of FLOPs as compared to BERT$_{\text{BASE}}$. $y$-axis is the relative drop in accuracy from the baseline. Pruning SOTA numbers are taken from Kwon et al. (2022), while distillation baselines are from Sanh et al. (2020); Sun et al. (2019). Right plot compares various techniques for MobileNetV3 compression on ImageNet-1K dataset. *MobileNetV3* corresponds to MobileNetV3 models with different width multiplier. *TuNAS, MorphNet* are SOTA techniques for scalable compression. TuNAS takes a blackbox approach to model compression, whereas MorphNet takes a more direct approach by optimizing FLOPs regularized objective.

these approaches are often computationally expensive as they take a blackbox view of the problem and perform combinatorial search over the space of architectures. Recent works have tried to open this blackbox to speed up the search process. One prominent line of work here is based on weight sharing which involves training a large surrogate network with many redundant operations to quickly evaluate the quality of an architecture in the search space (Liu et al., 2018; Shin et al., 2018; Cai et al., 2018; Tan et al., 2019). However, these techniques do not scale well to large search spaces, as they require storing a gigantic network. Despite recent advances such as TuNAS (Bender et al., 2020) for reducing the size of the network, these techniques can be an order of magnitude slower and less accurate than our proposed method (see Fig 1). See Section 2 for a thorough discussion on other related works.

**Whitebox Compression Techniques.** Among the category (b) techniques mentioned above, a prominent line of work has focused on unstructured pruning of weights with non-uniform budget allocation across layers (Han et al., 2015; Lin et al., 2020b; Renda et al., 2020; Kusupati et al., 2020). However, any gain in FLOPs using unstructured pruning is hard to translate to real latency gain as modern hardware – like GPUs, TPUs – are more geared towards dense matrix operations. So it is more fruitful to focus on structured building blocks such as neuron pruning, which removes entire neurons/channels, and low-rank factorization of weights, which is closely related to neuron pruning. Recent techniques in this line of work add a latency/FLOPs regularizer to the standard cross entropy loss (Gordon et al., 2018; Cai et al., 2018; Chaudhuri et al., 2020) to bias the model towards lower number of neurons. Unfortunately the resulting objective is discrete and difficult to optimize. To alleviate this, existing works have designed continuous surrogates that are more amenable to SGD style optimization. These methods either work in the space of probability distributions over pruned models and optimize the "expected objective" (Chaudhuri et al., 2020; Louizos et al., 2017; Wang et al., 2019) or replace the discontinuous FLOPs regularizer with a continuous surrogate such as $\ell_1$ norms of the weights of the network (Gordon et al., 2018). However, the former class of techniques are often unstable, hard to implement in practice, and empirical studies indicate that their performance is similar to that of simple magnitude based pruning (Gale et al., 2019) (also see left plot of Fig. 1). Furthermore, as we show in this work, the latter class of techniques fail to enforce sparsity in the presence of batch, layer normalization (see Section 3). Even in the absence of batch, layer normalization, these techniques require adhoc post-processing steps to output exact sparse solutions.

**Our Approach**: In this work, we propose a whitebox compression technique that addresses the above described optimization issues. Specifically, we propose a novel FLOPs/latency surrogate based on $\frac{\ell_1}{\ell_2}$ norm that works even in the presence of batchnorm, layernorm. Our approach applies to a large class of efficient building blocks – like unstructured sparsity, neuron pruning, quantization – for which we can express the FLOPs of the model with a $\frac{\ell_1}{\ell_2}$ surrogate (see Table 1). While our surrogates are continuous, they are non-differentiable. In such cases standard optimizers such as SGD, Adam can be quite slow to converge (Parikh et al., 2014). To overcome this, we propose a projection operation on the mask variables, after each SGD step. Our proposed method speeds up the convergence and also outputs *exact sparse solutions* thus eradicating need for post-hoc thresholding. Finally, our approach

is much faster than SOTA blackbox optimization techniques and runs in almost the same amount of time as single model training run.

We implement our algorithm with multiple building blocks including pruning, low-rank factorization, quantization, and apply it on multiple problems in the domain of image classification and NLP. In particular, we demonstrate the effectiveness of our technique for MobileNetV3 compression on ImageNet (see Fig. 1), where our method can learn an architecture with up to 15% (11%) lower FLOPs (latency) on Pixel 6 mobile phones, without any drop in accuracy. Here our approach is more accurate than MorphNet, a SOTA technique which focuses exclusively on neuron-pruning, as well as, TuNAS, a SOTA NAS technique. Furthermore, in terms of training time, our method is $3\times$ cheaper than TuNAS. We would like to highlight that MobileNetv3 is a highly optimized architecture found using efficient NAS techniques (Howard et al., 2019), and our technique is able to compress this architecture further.

One exciting application of our work is that we can apply it to optimize certain "foundational" baseline models for individual fine-tuning tasks. For example, for compression of BERT on GLUE benchmarks, our method achieved $40 - 50\%$ reduction in FLOPs with only $1\%$ drop in accuracy (see Fig 1). Moreover, our technique dominates standard model compression baselines. Similarly for smaller vision classification tasks, our technique compresses MobileNetV3, EfficientNet suite of architectures and identifies $1.2\times$-$1.4\times$ cheaper architectures without significant loss in accuracy (see Figure 4). Our technique also outperforms SOTA model compression techniques for ResNet by upto 1.5% on ImageNet (see Figure 3) We would like to note that all these results are obtained at almost the same cost as that of training a single model for the task. Finally, we also demonstrate the versatility of our method by using it to quantize a CNN on CIFAR-10, and learning optimal bit-widths for each of its layers. Our technique found a model that is 55% smaller than the baseline float-16 model, while achieving the same accuracy (see Figure 5). Here is a summary of our contributions:

**(1).** We provide an end-to-end neural network compression technique that directly optimizes the FLOPs regularized objective leading to compression during training. Our algorithm can be used with many popular efficient building blocks including pruning, low-rank factorization, quantization, and can optimize for on-device inference latency.

**(2).** We design a novel $\frac{\ell_1}{\ell_2}$ regularized surrogate for latency that works even in the presence of batchnorm, layernorm. Our algorithm is fast and runs in the same amount of time as single model training, and doesn't require any post-processing steps.

**(3).** We demonstrate the performance of our technique on both language and vision tasks. Moreover, for transfer learning settings where the goal is to take a baseline architecture and optimize it for individual tasks, our techniques outperform SOTA techniques in the broad-domain of automated neural compression.

## 2 RELATED WORK

### 2.1 NEURAL ARCHITECTURE SEARCH

Early works on NAS treated the problem as a purely blackbox optimization (BO) problem. These works relied on BO techniques such as random search (Li & Talwalkar, 2020), Gaussian process optimization (Kandasamy et al., 2018), and zeroth-order gradient descent (Tan et al., 2019; Zoph & Le, 2016), evolutionary algorithms to optimize the NAS objective and identify a good architecture. Several works have improved upon these algorithms using heuristics such as early stopping (Li & Talwalkar, 2020). Nonetheless, these techniques are computationally expensive, as evaluating the optimization objective at any point requires training a neural network from scratch. Moreover, due to computational complexity, these techniques perform a very coarse grained search and are not suited for fine-grained search over sparsity or low-rank structures.

**One-Shot NAS** - Recent works have tried to open the blackbox a bit. These techniques, termed as One-Shot NAS, aim to return the searched architecture as well as its optimal weights in a single pass. In these techniques, the search space is first transformed to the space of probability distributions over architectures. Next, a surrogate model is trained to quickly evaluate the optimization objective at any input (Bender et al., 2020; Liu et al., 2018; Pham et al., 2018; Mei et al., 2019; Chaudhuri et al., 2020). While these techniques are fast, they involve joint training of the surrogate model during the search process. This joint training often makes the optimization process unstable (Elsken et al., 2019). Since our method uses a gradient descent like paradigm, it sidesteps such issues. Further, prior work has shown evidence that such auxiliary models do not often correlate with the actual model performance

(Pourchot et al., 2020; Yu et al., 2019; Zela et al., 2020; Zhang et al., 2020b) in various settings.

**Zero-Cost Proxies** - There have also been techniques which look at data-independent zero-cost proxies for estimating the performance and latency of a network. These rely on proxy tasks(Wang et al., 2023; Li et al., 2021) to come up with an estimate of the actual performance. However, recent work has shown that simple baselines such as "number of parameters" and "FLOPs" are surprisingly competitive with all leading techniques (White et al., 2023). The main downsides of using zero-cost proxies are that they may be unreliable, especially on larger search spaces (White et al.; 2023). They also may have biases, such as preferring larger models (Ning et al., 2021) or wide channels (Chen et al., 2022). It has been shown that zero-cost proxies for CNNs do not transfer well to transformers(Zhou et al., 2022). In contrast, our method provides a simple regularizer and training recipe which can be applied to a wide range of base architectures and tasks, as we demonstrate in our experiments. We further refer the reader to a recent survey(White et al., 2023) for a more thorough view on the landscape of NAS.

**Hardware-aware NAS for Efficient ML** Several recent works at the intersection of efficient ML and NAS have realized the importance of explicitly accounting for the hardware in the search process (Tan et al., 2019; Chu et al., 2021; Lin et al., 2020a; Dong et al., 2021; Zhang et al., 2020a; Benmeziane et al., 2021; Cai et al., 2018). These works incorporate the actual inference time in their search objectives, instead of surrogates such as FLOPs. The inference time maybe estimated using another neural network (Łukasz Dudziak et al., 2021), or through latency tables for basic arithmetic operations on the target platform (Yang et al., 2018). Many of these works rely on greedy, random search heuristics to solve the resulting objective (Lin et al., 2020a; Dong et al., 2021). However, these heuristics either take a lot of time to find the optimal architecture or are not guaranteed to converge to an optimal solution. There are some works that rely on the NAS algorithms described above (Tan et al., 2019; Chu et al., 2021; Bender et al., 2020). However, these techniques face the same scalability and optimization issues as previously mentioned.

## 2.2 MODEL COMPRESSION

The field of model compression is vast. Here, we focus on techniques that perform training-time compression (as opposed to post-training compression) using the following building blocks: unstructured sparsity, pruning and low-rank factorization. Early works in unstructured sparsity and pruning relied on magnitude, gradient based pruning (Han et al., 2015; Frankle & Carbin, 2018; Gale et al., 2019). Several works have explored more sophisticated scoring metrics for pruning (Karnin, 1990; Molchanov et al., 2016; 2019; Guo et al., 2016; Dong et al., 2017). Other techniques include adding sparsity inducing norms such as $\ell_0, \ell_1$ to the training objective (Louizos et al., 2017; Kusupati et al., 2020; Tiwari et al., 2021). A number of works have also explored low-rank factorization for model compression (Jaderberg et al., 2014; Lu et al., 2016; Xu et al., 2019; Hsu et al., 2021). Some of these techniques again rely on sparsity inducing regularizers to induce the low-rank structure (Wang et al., 2019; Hsu et al., 2021). Others rely on SVD based pruning. Some recent works try and optimize FLOPs regularized objective to perform pruning, low-rank factorization (Gordon et al., 2018; Chaudhuri et al., 2020). However, as we discussed in the introduction, the resulting optimization techniques are often unstable and difficult to use in practice, in particular due to the large number of hyper-paramters needed by them. There have also been specialized methods developed for particular architecture types and modalities. Yu et al. (2022) present a unified compression framework for vision transformers, and Shi et al. (2023) present a similar pruning framework for multiple modalities. While our method is similar to these works, we note that our method can work across architecture types, modalities and training paradigms, and is agnostic to particular quirks of each of these domains.

## 3 METHOD

In this section, we describe our approach for model compression. For simplicity of presentation, we illustrate our technique on feed-forward networks and restrict ourselves to structured pruning. The ideas here can be extended to other architectures (*e.g.,* 1x1 convolutions in CNNs), and other efficient building blocks (*e.g.,* unstructured sparsity, low-rank factorization, quantization) in a straightforward manner (see Table 1 for details).

### 3.1 REGULARIZING THE FLOPS

Consider the following problem: we are given a pre-trained feed forward neural network (FFN) $f^*(x) = \sigma(W_D^* \sigma(W_{D-1}^* \sigma(\dots \sigma(W_1^* x))))$, where $W_i^* \in \mathbb{R}^{d_{i+1} \times d_i}$ for all $i \in [D]$, and a dataset

$\{(x_i, y_i)\}_{i=1}^n$. Our goal is to compress $f^*$ while simultaneously performing well on the learning task. This problem can be formulated as the following optimization problem

$$\min_{\mathcal{W}} \frac{1}{n} \sum_{i=1}^{n} \ell(x_i, y_i; \mathcal{W}) + \lambda \times \text{Latency}(\mathcal{W}). \tag{1}$$

Here $\mathcal{W} = \{W_i\}_{i=1}^D$, with $W_i \in \mathbb{R}^{d'_{i+1} \times d'_i}$ being the weight matrix at layer $i$, $\lambda$ is the regularization parameter which trades-off latency with accuracy and $\ell$ is the supervised loss. Directly optimizing the above objective is intractable because $\text{Latency}(\mathcal{W})$ is a discrete function of the dimensions of weight matrices, and is hardware specific.

We now present a technique for solving Equation (1). To begin with, we substitute $\text{Latency}(\mathcal{W})$ with $\text{FLOPs}(\mathcal{W})$[1]. In App B.1, we extend it to actual latency. The objective in this case is given by

$$\min_{\mathcal{W}} \frac{1}{n} \sum_{i=1}^{n} \ell(x_i, y_i; \mathcal{W}) + \lambda \sum_{i=1}^{D} d'_i d'_{i+1}. \tag{2}$$

To solve this objective, we associate masks with each neuron in the network. In particular, we parameterize the weight matrix in the $i^{th}$ layer as $W_i \times \text{diag}(\alpha_i)$. Here $\alpha_i \in \{0, 1\}^{d_i}$ are the mask variables of layer $i$. If $\alpha_{i,j}$ is set to 0, then the $j^{th}$ neuron in the $(i-1)^{th}$ layer will be pruned. The FLOPs regularizer[2] can now be written in terms of masks as $\sum_{i=1}^D \|\alpha_i\|_0 \|\alpha_{i+1}\|_0$, where $\alpha_{D+1}$ is the static vector of all 1's. To make this objective continuous and amenable to gradient based optimization, one class of techniques place a Bernoulli distribution $\text{Bern}(p_{i,j})$ over each of the masks $\alpha_{i,j}$ and solve the resulting smoothed objective where expectation is taken w.r.t. the random masks $\alpha_i$'s (Chaudhuri et al., 2020; Louizos et al., 2017; Wang et al., 2019) The resulting problem is NP-hard, and the discrete nature of $\alpha_i$'s makes the optimization unstable. To overcome this, Chaudhuri et al. (2020); Louizos et al. (2017); Wang et al. (2019) rely on a heuristic which involves relaxing Bernoulli distribution to a continuous distribution such as LogisticSigmoid. However, the main drawback of the resulting algorithm is that it is hard to implement in practice and requires very careful annealing of the parameters of LogisticSigmoid distribution. Further, the performance of such techniques is not well understood theoretically, even for simple and fundamental problems such as sparse linear regression.

Another common approach to convert the discrete objective in Equation (2) into a continuous function is to replace the $\ell_0$ norm on $\alpha_i$'s with $\ell_1$ norm

$$\min_{\mathcal{W}, \alpha_i \in \mathbb{R}^{d_i}} \frac{1}{n} \sum_{i=1}^{n} \ell(x_i, y_i; \alpha, \mathcal{W}) + \lambda \sum_{i=1}^{D} \|\alpha_i\|_1 \|\alpha_{i+1}\|_1. \tag{3}$$

This approach is both theoretically grounded (Tibshirani, 1996; Negahban et al., 2009) and easier to implement in practice (Parikh et al., 2014; Yun et al., 2021). Consequently, recent SOTA compression techniques relied on $\ell_1$ norm surrogates to compute the FLOPs regularizer (Gordon et al., 2018; Shi et al., 2023). A major drawback of $\ell_1$ norm though is that it does not promote sparsity in the presence of batch normalization and layer normalization (Ioffe & Szegedy, 2015; Ba et al., 2016). To see this, consider the following 1-hidden layer network: $\sigma(\text{BN}(W_2 \text{diag}(\alpha_2)\sigma(\text{BN}(W_1 \text{diag}(\alpha_1)x))))$. One can scale down all entries of $\alpha_1$ and scale up the weights $W_1$ without affecting the output of the network. Doing this reduces the objective value in Equation (3), but doesn't induce any sparsity in the network. In practice, we in fact notice this behaviour during optimization of Equation (3), which leads to sub-optimal solutions. We demonstrate this phenomenon empirically in Section 3.3. Note that adding $\ell_2$ penalty on the weights (*i.e.,* weight decay) doesn't mitigate this issue as any scaling of $\alpha's$ can be absorbed by the batch norm parameters without changing the output of the network. Further, such approaches also need a post-training thresholding step on the masks to achieve sparsity in practice, adding another hyper-parameter to the method.

---

[1]FLOPs is also a discrete function of dimensions of $W_i$, and the resulting optimization problem is still intractable.

[2]The expression we write here actually corresponds to the Multiply-Accumulate Operations (MACs). Each MAC usually corresponds to two FLOPs. However, we abuse notation slightly and use FLOPs throughout the paper, since this term is more widely used in prior literature.

## 3.2 Inducing sparsity through $\frac{\ell_1}{\ell_2}$ regularizer

We now introduce our approach for making the objective in Equation (2) continuous. Instead of using $\ell_1$ as a proxy, we replace $\ell_0$ norm over masks ($\|\alpha_i\|_0$) with $\frac{\ell_1}{\ell_2}$ penalty ($\sqrt{d_i}\|\alpha_i\|_1/\|\alpha_i\|_2$) and solve the following optimization problem

$$\min_{\mathcal{W}, \alpha_i \in \mathbb{R}^{d_i}} \frac{1}{n} \sum_{i=1}^{n} \ell(x_i, y_i; \alpha, \mathcal{W}) + \lambda \sum_{i=1}^{D} \frac{\sqrt{d_i}\|\alpha_i\|_1}{\|\alpha_i\|_2} \frac{\sqrt{d_{i+1}}\|\alpha_{i+1}\|_1}{\|\alpha_{i+1}\|_2}. \tag{4}$$

The $\sqrt{d_i}$ term in the numerator normalizes the penalty to lie between $[0, d_i]$. When $\alpha_i$'s are all 1's, the regularizer evaluates to FLOPs. Observe that this regularizer is invariant to scaling of $\alpha$'s. Consequently, the value of the regularizer cannot simply be reduced by scaling down $\alpha_i$'s. In our experiments in Sections 3.3 and 4.3, we show that this handles batch, layer normalizations better than $\ell_1$ regularizer. Several works have studied this regularizer in the context of sparse linear regression and showed that is recovers the underlying sparse signal under mild conditions on the data (Yin et al., 2014; Rahimi et al., 2019; Wang et al., 2020a). Yang et al. (2019) and Diao et al. (2023) used a similar $\frac{\ell_1}{\ell_2}$ regularizer for network pruning, but their techniques don't optimize latency or FLOPs, and rely on post-training thresholding to get sparsity.

For certain technical reasons described in the next paragraph, we add a positivity constraint on $\alpha_i$'s and solve the following objective

$$\min_{\mathcal{W}, \alpha_i \in \mathbb{R}_+^{d_i}} \frac{1}{n} \sum_{i=1}^{n} \ell(x_i, y_i; \alpha, \mathcal{W}) + \lambda \sum_{i=1}^{D} \frac{\sqrt{d_i}\sum_{j=1}^{d_i}\alpha_{i,j}}{\|\alpha_i\|_2} \frac{\sqrt{d_{i+1}}\sum_{j=1}^{d_{i+1}}\alpha_{i+1,j}}{\|\alpha_{i+1}\|_2}. \tag{5}$$

Note that we consider $\alpha_i \in \mathbb{R}_+^{d_i}$ rather than discrete or bounded values. We would like to highlight that this change doesn't reduce the representational power of our model. It is mainly done for computational reasons.

**Importance of positivity constraints.** The objective in Equation (4) is continuous, but not smooth. For such losses, standard optimization techniques such as SGD, Adam are slow to converge to stationary points (Boyd et al., 2004). Furthermore, these algorithms don't output exact sparse solutions. This forces additional post-processing steps to be introduced into the compression pipeline. For example, Gordon et al. (2018); Yang et al. (2019) rely on Adam optimizer and add a pruning step at the end, where masks that are close to 0 are pruned away. This is quite cumbersome in practice as one needs to choose appropriate thresholds for pruning, which introduces an additional tunable hyper-parameter, and needs re-training after pruning. To overcome this, we add a positivity constraint to the mask variables and modify the objective to Equation (5). This makes the regularizer smooth (except at all 0's vector), and easy to optimize using SGD, Adam. After each SGD/Adam update, we simply project the masks back to the space of positive real numbers. The overall update looks as follows

$$\mathcal{W} \leftarrow \mathcal{W} - \eta \nabla_{\mathcal{W}}(\mathcal{L}(\alpha, \mathcal{W}) + \lambda \mathcal{R}(\alpha)), \quad \alpha \leftarrow \max(0, \alpha - \eta \nabla_{\alpha}(\mathcal{L}(\alpha, \mathcal{W}) + \lambda \mathcal{R}(\alpha))).$$

Here $\mathcal{L}(\alpha, \mathcal{W})$ is the empirical risk and $\mathcal{R}(\alpha)$ is the regularizer. Notice, the only additional step compared to traditional optimization, is the clipping of $\alpha$'s. In our ablation studies in Sections 3.3 and 4.3, we validate the importance of this projection step, together with $\frac{\ell_1}{\ell_2}$ norm, in encouraging sparse solutions.

**Hardware-aware compression** - While we deal with FLOPs in this section, our method can also be extended to optimize the actual latency. We model the on-device latency as a sum of latencies of the individual matrix multiplications involved in the model. The latencies are looked up from an interpolated latency table constructed from on-device measurements. The $\frac{\ell_1}{\ell_2}$ regularizer is crucial to this interpolation, as it is normalized and lies between $[0, d_i]$. We refer the reader to Appendix B.1 for more details on our approach and empirical evaluation.

## 3.3 Verification of design choices

To empirically demonstrate the drawbacks of using $\ell_1$ penalty for model compression, we perform experiments on the FashionMNIST dataset with a single hidden layer fully connected network which

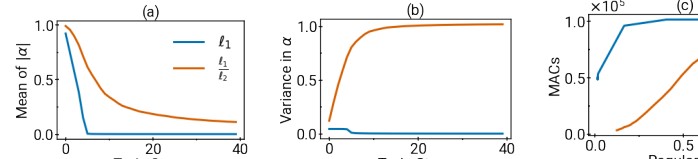

Figure 2: **Comparison of $\ell_1$, $\frac{\ell_1}{\ell_2}$ induced FLOPs regularizer for pruning on FashionMNIST**: Figures (a) and (b) depict the evolution of the statistics of the mask variables ($\alpha$) as training progresses. Figure (c) shows the relation between the actual FLOPs of the model and the value of the proxy computed by Equations 3, 4. Figure (d) shows the evolution of the Frobenius norm of the weight matrix.

has a batch norm layer after the first linear layer. We prune out the input to the network using a mask $\alpha$ on the input. We compare the performance of networks compressed using FLOPs regularizer induced by $\ell_1$ and $\frac{\ell_1}{\ell_2}$ norms. We use SGD for optimization of both the objectives. Furthermore, we pre-train the network using standard CE loss, and initialize $\alpha = \mathbf{1}$. We track the variance of the absolute values of the entries of $\alpha$, i.e. $\frac{\sum_{i=1}^{d}(|\alpha_i| - \mu_\alpha)^2}{d}$, where $\mu_\alpha = \frac{\sum_{i=1}^{d}|\alpha_i|}{d}$. We also track the mean $\mu_\alpha$ of the absolute values of the entries of $\alpha$. Finally, we plot out the curve between FLOPs and the considered norm of $\alpha$ (*i.e.*, $\ell_1$, $\frac{\ell_1}{\ell_2}$). Figure 2 presents the results from these experiments. We can see that the $\ell_1$ objective is mis-aligned with the actual value of FLOPs, while the regularizer computed using $\frac{\ell_1}{\ell_2}$ is a better proxy. We also find that the mean and variance of $\alpha$'s sharply decreases when $\ell_1$ induced FLOPs regularizer is used for compression. This indicates that all entries of $\alpha$ are uniformly scaled down to a small, non-zero value, reducing the value of the regularizer, while not providing any sparsity. As seen from the figure, $\frac{\ell_1}{\ell_2}$ does not suffer from this drawback. Finally, we note that the frobenius norm of the weight matrix $W$ increases when $\ell_1$ regularization is used on $\alpha$, suggesting that the network is simply scaling down $\alpha$'s and scaling up the weights to evade the regularizer.

## 4 EXPERIMENTS

In this section, we apply our framework to large scale pre-training and transfer learning tasks on standard language and vision benchmarks. To demonstrate the versatility of our technique, we perform experiments on multiple model families (MobileNet, EfficientNet, ResNet, BERT), and multiple building blocks (pruning, low-rank factorization). Note that in this section, we provide accuracy v/s MACs (Multiply-Accumulate operations) trade-off for various tasks[3]. Since we focus on *structured pruning*, a decrease in FLOPs (or MACs) would correlate with a decreased latency as well. In addition to this, in Sec 4.3, we also present experiments using the actual on-device latency instead of FLOPs and show that our searched models are indeed faster on device. Further, we also present a case study integrating quantization into our framework in Appendix A.1, demonstrating its versatility.

### 4.1 LARGE SCALE CLASSIFICATION ON IMAGENET

**MobileNet Family** - We begin by comparing the performance of our technique with baselines on MobileNetV3 compression, for ImageNet classification. We rely on low-rank factorization + pruning for the compression. The results from this experiment are presented in Figure 1. By varying the strength of our regularization, we obtain models with different MACs and accuracies. We find that models produced by our method significantly outperform MobileNetV3 and TuNAS in the high and mid-MACs regime. In particular, for the same accuracy as MobileNetV3Large, our approach finds a model with $15\%$ fewer MACs. In comparison with TuNAS, we achieve 30% reduction in MACs at the same level of accuracy. We however find that our model is at par with MobileNetV3Small in the low MACs regime, indicating that the former is already well-tuned for this task. In terms of compute needed for training, TuNAS is the most expensive among all the techniques we tried; it took 2 days to train with our hardware setup. In contrast, our method took 13 hours ($3 - 4\times$ faster than TuNAS), and MorphNet took 10 hours. Note that MobileNetv3 is a highly compressed model for edge deployment, and previous works have found it challenging to compress the model further. Our method can still provide a better FLOPs v/s accuracy trade-off, providing evidence for its efficacy.

---

[3]Note that while we use the term "FLOPs" to describe computational cost in the paper, we report MACs for computer vision models, in line with prior work.

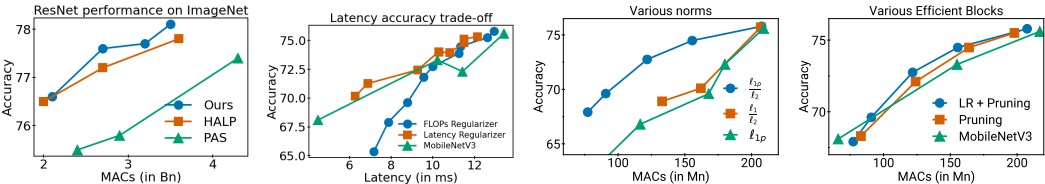

Figure 3: **(a) Pruning ResNet on ImageNet** - We compare against HALP and PAS, two recent SOTA techniques to prune ResNet-50, and achieve better performance over different FLOP regimes. **(b),(c),(d) - Ablation studies on Mobilenetv3** We compare using $\ell_1$ and $\frac{\ell_1}{\ell_2}$ norms in our regularizer, with subscript $p$ indicating that projected-Adam was used for optimization. We also experiment with combining low-rank (LR) factorization with channel pruning. Finally, we show on device latency-accuracy tradeoff with using the actual latency regularizer for compressing MobileNetv3

**ResNet Family** - We also compress the ResNet architecture for ImageNet classification, using our method. In particular, we compress the $1 \times 1$ convolutions using pruning and low-rank factorization. We compare our method against HALP (Shen et al., 2021) and PAS (Li et al., 2022), two state of the art methods for neural architecture search and model compression for ResNet. Our method compresses ResNet-101 to a model with similar FLOPs as ResNet-50, while simultaneously achieving better performance than the baseline ResNet-50. Furthermore, our technique outperformns SOTA methods for the same number of FLOPs by up to 1.5%, as seen in Fig 3.

## 4.2 TRANSFER LEARNING

A common paradigm in deploying machine learning models today is to first pre-train them on a large scale dataset such as ImageNet, and then fine-tune them for the desired target task. However, deploying large models is not feasible on edge devices. Our technique provides a light-weight modification to the standard fine-tuning procedure by producing a highly compressed model with comparable transfer learning performance on the specific task. We demonstrate this on vision and language tasks.

**Vision tasks.** We consider the task of fine-tuning an ImageNet pre-trained model for a smaller dataset. We consider Cars196 (Krause et al., 2013) and Food101 (Bossard et al., 2014) as the target datasets, and compare against the MobileNetV3 and EfficientNet families of models. We use ImageNet pre-trained models for initialization. We plot the FLOP-accuracy curves in Fig 4. We compress MobileNetv3Large and EfficientNet-B4 and EfficientNet-B2 architectures while fine-tuning them on the target target task. We find that our method consistently improves over baseline architectures across various FLOPs regimes. This is because our technique is able to adaptively prune the model based on the difficulty of the classification task. On both the tasks, we see 1% accuracy gains over MobileNetV3 small. The accuracy gains persist at the latency footprint of MobileNetV3Large-0.75, where we see over 1.5% accuracy gains on both datasets. On EfficientNet, we see upto 40% reduction in FLOPs without any drop in accuracy on Food101, and around 20% reduction in FLOPs on the Cars196 dataset for the largest models (B4). We also see around 30% FLOP reduction while maintaining the transfer learning performance of the B1 and B0 variants. This demonstrates that our learnt models can scale better than the heuristic scaling described in (Tan & Le, 2019).

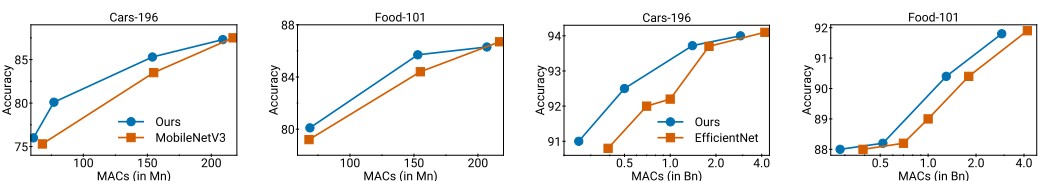

Figure 4: **Accuracy-FLOPs trade-off on vision transfer learning tasks**: Figures (a) and (b) depict the the fine-tuning performance of models found by our method while compressing MobileNetv3Large and baseline MobileNetV3 on Cars-196 and Food-101 datasets. Figures (c) and (d) show the performance on the EfficientNet family of architectures, where baselines are EfficientNetB0-B4, while our method compresses EfficientNet B4 and B2.

**Fine-tuning BERT on GLUE.** We consider 5 datasets of the GLUE benchmark (Wang et al., 2018) that are commonly used in the literature, and fine-tune a pre-trained BERT-Base model with our FLOPs regularizer. We re-parameterize the weight matrices of the feed forward network of each transformer block with our low-rank+sparse parameterization. We compare our approach against model pruning, where SOTA numbers are taken from Fig. 6 of Kwon et al. (2022), reporting the maximum accuracy among Liu et al. (2021); Lin et al. (2020c); Lagunas et al. (2021); Wang et al. (2020b); Xia et al. (2022); Sajjad et al. (2023). We also report the performance of widely-used distillation based baselines (Sanh et al., 2020; Sun et al., 2019). Figure 1 presents the average performance on the 5 datasets, and Figure 7 in appendix presents the performance on each dataset. In both these figures, we plot the relative non-embedding FLOPs of the compressed model w.r.t BERT-base against the drop in accuracy w.r.t BERT-base (similar to Kwon et al. (2022)). We find that on 4 of the 5 datasets considered, our technique provides a higher accuracy for the same number of FLOPs, indicating the efficacy of our method. On MRPC, a dataset with very few samples, our method is worse off for models with higher FLOPs, but outperforms the baselines in the low FLOP regime.

### 4.3 Ablation Studies on MobileNetV3

**Effect of optimization choices.** In section 3 we provided small scale experiments to justify our design choices of using projected-Adam and $\frac{\ell_1}{\ell_2}$ norm. In this section we perform large-scale ablation studies on MobileNetV3 for ImageNet training. The results from this experiment are presented in Figure 3. Without projected-Adam, we notice that the optimization algorithm doesn't converge to sparse solutions. Consequently, the resulting models do not have large reduction in MACs. The accuracy of these models also takes a big hit. On the other hand, using $\ell_1$ norm based FLOPs regularizer with projected-Adam suffers from the scaling issue described in Sec 3.3. This leads to a large fraction of channels being pruned for some blocks, producing a model with deteriorated accuracy. Our method has 2-4% better accuracy in the high and mid FLOPs regimes than these alternatives.

**Comparing different building blocks.** In Table 1, we described ways to integrate various building blocks into our framework. In Figure 3, we demonstrate the accuracy *vs* inference time trade-offs of using two of these building blocks in our framework, namely Pruning and Pruning+Low-rank Factorization. We find that the extra flexibility provided by the Low-Rank Factorization leads to models with fewer MACs for the same accuracy, and the difference is even more pronounced for smaller models. We note that channel pruning alone can give us 10% reduction in MACs over the MobileNetV3 family at the same accuracy level. In particular, at $73.4\%$ accuracy, our model has 136Mn MACs compared to 155Mn MACs of the MobileNetV3 family model. Similarly, at $75.5\%$ accuracy, our model has 198Mn MACs compared to 216Mn MACs of MobileNetV3 family model. Adding Low-Rank structure introduces another 5% reduction in MACs, with no loss in accuracy. This also shows the effectiveness of our algorithm across multiple building blocks.

**Hardware-aware compression.** We now optimize for actual on-deive latency by considering latency based $\ell_1/\ell_2$ surrogates (see Eq 7 in Appendix for more details on the surrogate). We provide empirical evidence on the effectiveness of this approach for MobileNetV3 on Pixel 6. We measure the latency on the device's CPU. We compare the accuracy-latency curves of models produced using FLOPs, latency regularizers (see Fig 3). Observe that using the latency regularizer leads to models with smaller latencies and consequently better latency-accuracy tradeoff compared to using the FLOP regularizer. We also find these models to have better performance than MobileNetV3 ($0.5 - 2\%$ improvement in accuracy for similar latency), despite MobileNetV3 being hand-crafted for faster inference on mobile devices. Note that latencies here are actual on-device inference latencies of the models.

## 5 Conclusion and Future Work

In this work, we presented an end-to-end technique for neural network compression. Our approach applies to a wide variety of efficient blocks including pruning, unstructured sparsity, quantization. At the core of our algorithm is a novel surrogate for FLOPs, latency that relies on $\frac{\ell_1}{\ell_2}$ norms, and works with batchnorm, layernorm. Our algorithm is computationally efficient and runs in same amount of time as needed for training a single model. We demonstrated the efficacy of our approach on various pre-training and transfer learning tasks on standard language and vision benchmarks. As a future work, it will useful to incorporate more efficient building blocks such as block diagonal matrices into our framework. Another interesting direction would be to make our technique more hardware aware by incorporating hardware level parameters such as tiling into our search process.

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

# A  USING QUANTIZATION WITH OUR FRAMEWORK

In this section, we present the parameterization of $W_i$ for quantization. Similar to the main paper, we consider a FFN. For each layer of the network, we would like to search over $\{1, 2, 4 \ldots B\}$ bit quantizations of its weights[4]. Let $W_i$ be the weight matrix of layer $i$. Let $\text{clip}(W_i; r_{i,l}, r_{i,u})$ be the weight matrix restricted to $[r_{i,l}, r_{i,u}]$

$$\text{clip}(W_i; r_{i,l}, r_{i,u}) = r_{i,u} - \text{ReLU}(r_{i,u} - r_{i,l} - \text{ReLU}(W_i - r_{i,l})).$$

When clear from the context, we use the short hand notation $\text{clip}(W_i)$ to denote $\text{clip}(W_i; r_{i,l}, r_{i,u})$. Let $W_{i,b}$ to be the $b$-bit quantization of $W_i$, and let $r_{i,l}^{(b)}, r_{i,u}^{(b)}$ be the range parameters associated with $W_{i,b}$. $W_{i,b}$ is obtained by uniformly gridding the range $(r_{i,u}^{(b)} - r_{i,l}^{(b)})$ into $2^b$ points and assigning each element of $W_i$ to its nearest grid point

$$W_{i,b} = r_{i,l}^{(b)} + \frac{r_{i,u}^{(b)} - r_{i,l}^{(b)}}{2^b - 1} \left\lfloor \frac{\text{clip}(W_i) - r_{i,l}^{(b)}}{(r_{i,u}^{(b)} - r_{i,l}^{(b)})/(2^b - 1)} \right\rceil.$$

Here $\lfloor \cdot \rceil$ denotes the round-to-nearest-integer function. To choose between $W_{i,1}, W_{i,2} \ldots W_{i,B}$, we introduce binary mask variables $\alpha_{i,1}, \alpha_{i,2} \ldots \alpha_{i,B}$. This leads us to the following parameterization of $W_i$

$$\alpha_{i,1} \left( W_{i,1} + \alpha_{i,2} \left( W_{i,2} - W_{i,1} + \alpha_{i,4} \left( W_{i,4} - W_{i,2} + \alpha_{i,8} \left( \ldots \right) \right) \right) \right) \tag{6}$$

$\alpha_{i,b} = 0$ implies the weights can be parameterized with fewer than $b$ bits. Observe that the above expression can be rewritten as

$$\alpha_{i,1}(1 - \alpha_{i,2})W_{i,1} + \alpha_{i,1}\alpha_{i,2}(1 - \alpha_{i,4})W_{i,2} + \alpha_{i,1}\alpha_{i,2}\alpha_{i,4}(1 - \alpha_{i,8})W_{i,4} \ldots$$

The FLOPs needed to compute the output of this layer is given by

$$\left[ \|\alpha_{i,1}(1 - \alpha_{i,2})\|_0 + 2\|\alpha_{i,1}\alpha_{i,2}(1 - \alpha_{i,4})\|_0 + 4\|\alpha_{i,1}\alpha_{i,2}\alpha_{i,4}(1 - \alpha_{i,8})\|_0 + \ldots \right] d_i d_{i+1}$$

Since searching for binary masks is computationally intractable, we make them continuous; that is, we let $\alpha_{i,b} \in [0, 1], \forall b \in \{1, 2, 4, \ldots B\}$. We consider a continuous FLOPs surrogate which is obtained by computing $\ell_1/\ell_2$ norm of

$$[\alpha_{i,1}(1 - \alpha_{i,2}), 2\alpha_{i,1}\alpha_{i,2}(1 - \alpha_{i,4}), 4\alpha_{i,1}\alpha_{i,2}\alpha_{i,4}(1 - \alpha_{i,8}) \ldots].$$

This leads us the following regularizer

$$\frac{\alpha_{i,1}(1 - \alpha_{i,2}) + 2\alpha_{i,1}\alpha_{i,2}(1 - \alpha_{i,4}) + 4\alpha_{i,1}\alpha_{i,2}\alpha_{i,4}(1 - \alpha_{i,8}) \ldots}{\sqrt{(\alpha_{i,1}(1 - \alpha_{i,2}))^2 + (2\alpha_{i,1}\alpha_{i,2}(1 - \alpha_{i,4}))^2 + (4\alpha_{i,1}\alpha_{i,2}\alpha_{i,4}(1 - \alpha_{i,8}))^2 \ldots}} d_i d_{i+1}.$$

**Remark 1.** *There are several other works that have attempted to learn the amount of quantization/precision to use at each layer Chen et al. (2021); Uhlich et al. (2019); Van Baalen et al. (2020). However, unlike our work, these works do not directly optimize for FLOPs, latency. We would like to note that our parameterization is closely related to the parameterization of Van Baalen et al. (2020).*

**Straight Through Estimator (STE).**  Note that the training objective for quantization is non-differentiable. So, in our experiments, we use STE to optimize the objective Bengio et al. (2013). This is a standard technique for performing quantization aware training.

## A.1  EXPERIMENTAL RESULTS

In this set of experiments, we consider CIFAR-10 classification and compress a 3 layer CNN using quantization. We use the quantization formulation presented in Table 1 and search over $\{2, 4, 8, 16\}$ bit quantizations for each layer. We compare with a baseline which uses the same level of quantization at each layer. Fig 5 presents the results from this experiments. The details of the implementation can be found in the appendix. We find that our technique compresses the model size by almost 55% without drop in accuracy (as compared to a model with 16-bit weights). Our technique also outputs a model which is 1.4% more accurate than a 2-bit quantized model with only 4% more FLOPs. In the plot on the right in Fig 5, we visualize the learned bit-widths of our models. We find that later layers are assigned a smaller bit width, indicating the importance of learning expressive filters early in the network. The different models in our plots were found by varying the the value of the regularizer coefficient, and hence no combinatorial search over bit-widths is required.

---

[4]$B$ is typically a power of 2

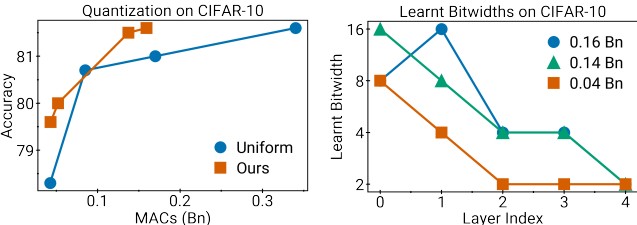

Figure 5: **Quantization on CIFAR-10**: Figure (a) compares the performance of our technique for dynamic quantization against fixed-bit quantization for a 4 layer CNN on CIFAR-10. The baselines have weights quantized to 2,4,8, 16 bits. Fig (b) depicts the learnt bitwidths for different layers of the models found by our technique, with the labels denoting the number of MACs (in Bn) of the models.

## B  HARDWARE AWARE COMPRESSION

### B.1  METHOD

In this section, we extend the FLOPs regularizer to take the latency on the target hardware into account. The resulting regularizer is especially useful for performing hardware aware network compression. Our key observation is that the inference on a neural network can be broken down into a series of matrix multiplication operations. For example, inference on a depth $D$ FFN involves $D$ matrix-vector multiplications, which take-up majority of the time. So, getting a good estimate of the inference time of the overall network boils down to having a good estimate of the latency of matrix-vector multiplication. To this end, we rely on lookup tables. Before the start of the pruning phase, we construct a 2-dimensional lookup-table $T$ whose $(d_1, d_2)^{th}$ entry is the on-device latency of multiplying a matrix of size $d_1 \times d_2$ with a vector of size $d_2$. Such a table is easy to construct, given access to the target device. Next, to incorporate the look-up table $T$ into our pruning algorithm, we convert it into a continuous function by performing linear interpolation on the entries in the table Späth (1995). To be precise, for any $(x, y) \in [d_1, d_1+1] \times [d_2, d_2+1]$, where $d_1, d_2 \in \mathbb{N} \cup \{0\}$, we define $T(x, y)$ as: $T(x, y) = t_1 + (t_2 - t_1)(y - d_2)$, where $t_1 = T(d_1, d_2) + (T(d_1 + 1, d_2) - T(d_1, d_2))(x - d_1)$, and $t_2 = T(d_1, d_2+1) + (T(d_1+1, d_2+1) - T(d_1, d_2+1))(x - d_1)$. Note that in contrast to black-box NAS techniques like Yang et al. (2018) which search over a discrete space of number of filters for each block, our approach needs the latency surrogate to be differentiable, and hence we need interpolated latency tables. See the appendix for details on how we construct the tables.

We use this interpolated lookup table to construct our *latency* regularizer as follows

$$\sum_{i=1}^{D} T\left( \frac{\sqrt{d_i}\|\alpha_i\|_{1p}}{\|\alpha_i\|_2}, \frac{\sqrt{d_{i+1}}\|\alpha_{i+1}\|_{1p}}{\|\alpha_{i+1}\|_2} \right). \tag{7}$$

In the above expression, our differentiable surrogate for $\|\alpha_i\|_0$ (*i.e.*, $\sqrt{d_i}\|\alpha_i\|_{1p}/\|\alpha_i\|_2$), is used to index the lookup table. We note that $\frac{\ell_1}{\ell_2}$ norm is very crucial for this technique to be successful. This is because $\frac{\sqrt{d_i}\|\alpha_i\|_{1p}}{\|\alpha_i\|_2}$ is normalized and always lies between $[0, d_i]$. In contrast, using $\ell_1$ norm surrogate in the regularizer gives us $T(\|\alpha_i\|_1, \|\alpha_{i+1}\|_1)$. Scaling $\alpha_i$ by a constant can drastically change this regularizer, and makes the optimization unstable.

### B.2  EXPERIMENTAL RESULTS.

In Eq 7, we propose a latency surrogate for optimizing the actual on-device inference latency. In this section, we provide empirical evidence of the effectiveness of this approach for MobileNetv3 on Pixel 6. We use CPU with no quantization. We compare the accuracy-latency curves of models produced using FLOPs, latency regularizers (see Fig 6). Observe that using the latency regularizer leads to models with smaller latencies and consequently better latency-accuracy tradeoff compared to using the FLOP regularizer. We also find these models to have better performance than MobileNetV3 ($0.5 - 2\%$ improvement in accuracy for similar latency), despite MobileNetv3 being hand-crafted for faster inference on mobile devices. Note that latencies here are actual on-device inference latencies of the models.

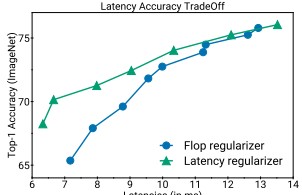

| Method | Latencies (ms) | Top-1 Accuracy | MACs (mn) |
|---|---|---|---|
| Latency Regularizer | 10.34 | 74.03 | 152.8 |
| | 12.1 | 75.25 | 184 |
| | 13.52 | 76.05 | 216.9 |
| MobileNetv3 | 10.22 | 73.3 | 155 |
| | 11.42 | 72.3 | 209 |
| | 13.41 | 75.6 | 217 |

Figure 6: Left plot shows the accuracy-latency curves of models obtained using FLOPs, latency regularizers. Right table compares the performance of our latency regularized models with MobileNetV3 baseline.

## C   IMPLEMENTATION AND EXPERIMENTAL DETAILS

In this section, we provide additional details about the implementation of our technique. We warm-start our pruning procedure with the pre-trained model (*i.e., $f^*$* in Section 3) that is made available to us. In our experiments, we noticed that this speeds up the convergence of our algorithm. For both MobileNetV3 and BERT compression, we rely on simultaneous pruning, low-rank factorization of weights (see Table 1 for details). Here, we parameterize weights $W_i$ as $U_i \text{diag}(\beta_i)V_i \text{diag}(\alpha_i)$; setting entries of $\beta_i$ to 0 helps reduce the rank of the weight matrix, and $\alpha_i$ helps in pruning. We initialize $U_i, V_i, \beta_i$ by performing SVD on the weight matrices of the pre-trained network. In our experiments, we apply our technique only to the $1 \times 1$ convolution layers in the network, for which the formulation of our regularizer remains the same as the one described in the preceding text. We anneal our regularization strength $\lambda$, increasing it linearly in order to stabilize the training. Finally, we fine-tune the model returned by our pruning algorithm on the training data to improve its performance (this simply involves setting the FLOPs regularization coefficient to 0). During the fine-tuning and pruning phases, we leverage the pre-trained model by adding a distillation loss to the standard cross-entropy loss Hinton et al. (2015). We perform distillation between the logits of the pre-trained model and the logits of the model being fine-tuned.

### C.1   IMAGENET PRETRAINING

Our algorithm was implemented using TensorFlow 2.0. We use the pre-trained MobileNetV3 (or EfficientNet) models provided in this framework to warm-start our models. We initialize $U_i, \beta_i, V_i$ to the SVD of the 1x1 convolution filters, and the entries of $\alpha_i$ uniformly at random between $[0, 0.5]$. We use Adam for optimization with its default parameters, and searched over learning rates in the set $\{10^{-4}, 5 \times 10^{-5}, 10^{-5}\}$, with cosine decay, which is standard practice. The distillation coefficient was searched among $\{0.1, 0.25, 0.5, 0.9\}$ and the distillation temperature was searched among $\{2, 3, 4\}$. For our ImageNet experiments, We trained our model for 70000 steps, linearly annealing the regularizer for the first 50000 steps. We fine-tuned the obtained model for another 50000 steps. For the transfer learning experiments, we reduced these to 25000 for training and 15000 for fine-tuning. We used a batch size of 2048 for all experiments. Our regularizer coefficient was varied from $10^{-8}$ to $10^{-6}$. This range was determined by looking at the magnitude of the cross-entropy loss and the FLOPs regularizer, and making sure that they are similar. MobileNet pre-training took up around 13 hours.

### C.2   TRANSFER LEARNING

**BERT.**   For the BERT fine-tuning experiments, we start off with a pre-trained BERT model and introduce our parameterization in a similar manner as described above. We use AdamW to optimize, and search over learning rates among $\{10^{-4}, 5 \times 10^{-5}, 10^{-5}\}$. Our regularizer coefficient was varied from $10^{-7}$ to $5 * 10^{-6}$. Each fine-tuning run taking between 20 mins - 1 hour.

**EfficientNet, MobileNet.**   For EfficientNet and MobileNet experiments, we have similar experimental setup and hyper-parameter search space as MobileNet ImageNet pretraining described in App C.1, with the exception that we do not do any model distillation. We also use RMSProp for EfficientNet with exponential decay as the LR schedule, as this was the optimizer of choice for its pre-training. We train for 25000 steps with the regularizer, and fine-tune for another 25000 steps.

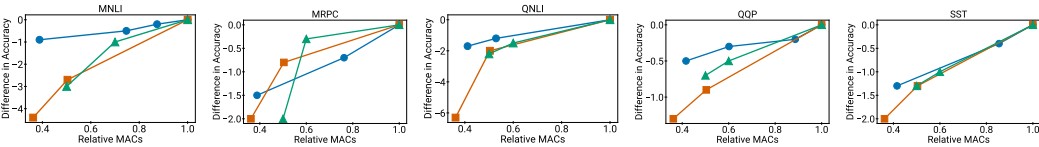

Figure 7: Fine-tuning tradeoffs of BERT on GLUE benchmark.
.

## C.3 QUANTIZATION

We train a CNN with four convolution layers, with [64,128,256,128] filters and kernel size of 3 with stride being 1 for each layer. We additionally have batch-norm layers after each conv layer. We search for learning rate over {1e-4, 5e-4, 1e-3, 5e-3} for the baseline and our model, and regularizer coefficient over {1e-9, 3e-9, 5e-9, 7e-9, 1e-8}. We train for 100 epochs with a batch size of 512 on a single V100 GPU, and use Adam with CosineDecay for the learning rate.

## C.4 LATENCY TABLES

As mentioned in the main paper, the actual on-device latencies are calculated on Pixel6 for our latency experiments. We populated the latency lookup-table $T$ specified in Appendix B.2 by profiling the corresponding $1 \times 1$ convolution/matrix-vector multiplication latency, on the device. Note that the convolution operation is much better optimized than matrix multiplication operation on the Pixel6 kernel. Hence, for our latency experiments on MobileNet, the latency table was populated by profiling the $1 \times 1$ convolution operations.

A 1x1 convoultion operation is identified by input dimension, input channels and the number of filters (output channels). Strides can also be different but all 1x1 convolutions in the MobileNet architecture have stride 1. In the MobileNet architecture we encounter feature maps with input dimensions $in_{dim} \in I = \{1, 7, 14, 28, 56, 112, 224\}$. Moreover, the input ($in_c$) and output channels ($out_c$) are constrained by $in_c, out_c \in D = \{d \mid \forall d \in \mathbb{N} \text{ and } d < 1281\}$. Hence we construct the table $T$, each member of which can be accessed via $T(in_{dim}, in_c, out_c)$. Note that profiling $T(in_{dim}, in_c, out_c)$. for every possible value of $(in_{dim}, in_c, out_c) \in I \times D \times D$ is expensive. We must therefore pick certain tuples $(in_c, out_c)$ for each $in_{dim} \in I$ for which we calculate actual on-device latencies. The rest of the table is populated using linear interpolation. We pick these tuples such that they cover the $1 \times 1$ convolutions that are encountered in the MobileNet Architecture. For $in_{dim} = \alpha$, let $\beta$ denote the maximum possible value of $in_c$, and $\gamma$ denote the maximum possible value of $out_c$ in MobileNet. We construct set $P_{in}$ which denotes values that are likely to be encountered by the regularizer for $in_c$ and similarly $P_{out}$ for $out_c$. Finally, the actual on-device latencies are calculated for $T(\alpha, P_{in} \times P_{out})$. Construction of $P_{in}$ and $P_{out}$ is done by choosing an appropriate $\theta$ and adding all values in the range $(\beta - \theta, \beta]$ to $P_{in}$, and $(\gamma - \theta, \gamma]$ to $P_{out}$. Also, from the remaining ranges i.e. $(0, \beta - \theta]$ and $(0, \gamma - \theta]$ points are sampled exponentially by choosing the midpoint of the range every time and changing the lower limit of the range to the midpoint for certain iterations.

The experimental setting and hyper-parameter configurations we use for latency table experiments is same as the one for FLOPs experiments (see Section C.1).

## D COMBINATION OF BUILDING BLOCKS

Table 1 presents the parameterization of weight matrices that lets us search over multiple building blocks simultaneously.

## E LIMITATIONS AND BROADER IMPACT

One limitation of our work is that we only study popular building blocks such as sparsity, pruning, low-rank factorization and quantization. Extending our work to more building blocks such as block sparsity and other forms of structured sparsity is an interesting future direction. Another limitation, which is related to the implementation of our technique, is the need to manually implement the

Table 1: Table describing regularizers used by our technique for various efficient building blocks . One can easily design regularizers for searching over a combination of building blocks. For example, row 5 presents regularizer for low-rank + pruning, which we use in our large-scale experiments.

| Efficient Building Block | Parameterization of $W_i$ | FLOPs ($i^{th}$ layer) | Regularizer (FLOPs surrogate) ($i^{th}$ layer) |
|---|---|---|---|
| Pruning | $W_i \times \text{diag}(\alpha_i)$ | $\|\alpha_i\|_0\|\alpha_{i+1}\|_0$ | $\frac{\sqrt{d_i}\|\alpha_i\|_{1p}}{\|\alpha_i\|_2} \frac{\sqrt{d_{i+1}}\|\alpha_{i+1}\|_{1p}}{\|\alpha_{i+1}\|_2}$ |
| Unstructured Sparsity | $W_i \odot \alpha_i$, where $\alpha_i \in \mathbb{R}_+^{d_{i+1}\times d_i}$, $\odot$ is the elementwise multiplication operator | $\|\text{Vec}(\alpha_i)\|_0$ | $\frac{\sqrt{d_i d_{i+1}}\|\text{Vec}(\alpha_i)\|_{1p}}{\|\text{Vec}(\alpha_i)\|_2}$ |
| Low-rank Factorization | $U_i\text{diag}(\beta_i)V_i$, where $U_i \in \mathbb{R}^{d_{i+1}\times d_{i,*}}$, $d_{i,*}=\min\{d_i, d_{i+1}\}$ | $(d_i + d_{i+1})\|\beta_i\|_0$ | $(d_i + d_{i+1})\frac{\sqrt{d_{i,*}}\|\beta_i\|_{1p}}{\|\beta_i\|_2}$ |
| Quantization (1, 2, 4 bit quantization) | $W_{i,1} + \alpha_{i,2}(\Delta_{i,2} + \alpha_{i,4}(\Delta_{i,4}))$, where $\alpha_{i,2}, \alpha_{i,4} \in [0,1]$, are mask variables, $W_{i,b}$ is the $b$-bit quantization of $W_i$, $\Delta_{i,2} = W_{i,2} - W_{i,1}$, $\Delta_{i,4} = W_{i,4} - W_{i,2}$ | $\|(1-\alpha_{i,2})\|_0 d_i d_{i+1} +$ $2\|\alpha_{i,2}(1-\alpha_{i,4})\|_0 d_i d_{i+1} +$ $4\|\alpha_{i,2}\alpha_{i,4}\|_0 d_i d_{i+1}$ | $\frac{\ell_1}{\ell_2}$ norm over the vector $[(1-\alpha_{i,2})$ , $2\alpha_{i,2}(1-\alpha_{i,4})$ , $4\alpha_{i,2}\alpha_{i,4}]$ $\times d_i d_{i+1}$ |
| Pruning + Low-rank Factorization | $U_i\text{diag}(\beta_i)V_i\text{diag}(\alpha_i)$, where $U_i \in \mathbb{R}^{d_{i+1}\times d_{i,*}}$, $d_{i,*}=\min\{d_i, d_{i+1}\}$ | $(\|\alpha_i\|_0 + \|\alpha_{i+1}\|_0)\|\beta_i\|_0$ | $\left(\frac{\sqrt{d_i}\|\alpha_i\|_{1p}}{\|\alpha_i\|_2} + \frac{\sqrt{d_{i+1}}\|\alpha_{i+1}\|_{1p}}{\|\alpha_{i+1}\|_2}\right)$ $\times \frac{\sqrt{d_{i,*}}\|\beta_i\|_{1p}}{\|\beta_i\|_2}$ |
| Pruning + Unstructured Sparsity | $(W_i \odot \beta_i) \times \text{diag}(\alpha_i)$, $\odot$ is the elementwise multiplication operator | $\|\text{Vec}(\beta_i \times \text{diag}(\alpha_i))\|_0$ | $\frac{\sqrt{d_i d_{i+1}}\|\text{Vec}(\beta_i \times \text{diag}(\alpha_i))\|_{1p}}{\|\text{Vec}(\beta_i \times \text{diag}(\alpha_i))\|_2}$ |
| Pruning + Quantization (1, 2, 4 bit quantization) | $\left(W_{i,1} + \alpha_{i,2}(\Delta_{i,2} + \alpha_{i,4}(\Delta_{i,4}))\right)$ $\times \text{diag}(\beta_i)$, | $\left(\|(1-\alpha_{i,2})\|_0 +\right.$ $2\|\alpha_{i,2}(1-\alpha_{i,4})\|_0 +$ $\left.4\|\alpha_{i,2}\alpha_{i,4}\|_0\right) \times \|\beta_i\|_0\|\beta_{i+1}\|_0$ | $\left(\frac{\ell_1}{\ell_2}\right.$ norm of the vector $[(1-\alpha_{i,2})$ , $\left.2\alpha_{i,2}(1-\alpha_{i,4}), 4\alpha_{i,2}\alpha_{i,4}]\right)$ $\times \frac{\sqrt{d_i}\|\beta_i\|_{1p}}{\|\beta_i\|_2} \frac{\sqrt{d_{i+1}}\|\beta_{i+1}\|_{1p}}{\|\beta_i\|_2}$ |

**FLOPs regularizer for various architectures.** An automated solution that takes in any architecture and computes the FLOPs regularizer would make our framework easy to use.

In terms of broader impact, we believe our technique can be used to find more efficient architectures for large language models such as GPT. This can help democratize these models, and also reduce their carbon footprint.

