# OpenReview forum: "End-to-End Neural Network Compression via $\frac{\ell_1}{\ell_2}$  Regularized Latency Surrogates"
_ICLR.cc/2024/Conference — Submitted to ICLR 2024_

### Official Review · Reviewer_qwYJ · 2023-10-31

**Soundness:** 3 good
**Presentation:** 3 good
**Contribution:** 2 fair
**Rating:** 5
**Confidence:** 5

**Summary:**

The paper introduces on model compression method, which is with low complexity, compared to the typical NAS.
The experiment shows that the 50% reduction is only with 1% performance loss. Besides, the author introduces its procedure of the math derivation.

But, such method is not novel enough, and similar idea (optimize the mask) has been widely discussed.
Secondly, model compression and quantization should be widely experimented in diverse models and modules.

**Strengths:**

The experiment shows that the 50% reduction is only with 1% performance loss. Besides, the author introduces its procedure of the math derivation.
The paper is with a well-written, and clearly present the contents.

**Weaknesses:**

But, such method is not novel enough, and similar idea (optimize the mask) has been widely discussed.
Secondly, model compression and quantization should be widely experimented in diverse models and modules, e.g., conv, transformer, linear.

**Questions:**

One question that arises is: what is the relationship between model compression and transfer learning? Do they occur in the same pipeline? For instance, is model compression done within the transfer learning process? Or is model compression performed first, followed by transfer learning to enhance performance?

---

> ### Author Response · Authors · 2023-11-18
> **Response to reviewer qwYJ**
>
> We thank the reviewer for the valuable feedback. Below we address some of the key concerns raised.
>
> $\textbf{Limited Novelty}$:  We would like to emphasize that introducing masks in the objective is not our contribution. Our main contribution is to address the challenges that arise in optimization of the resulting objective. Below we highlight our key contributions
>
> > Our first contribution is to design a novel $\ell_1/\ell_2$ regularizer for optimizing the FLOP regularized objective. This helps us effectively compress models with batchnorm and layernorms. This is in contrast to prior works which use $\ell_1$ based FLOPs regularizer and as a result fail to work well in the presence of layer/batchnorms. Please see  Figure 2 in our paper for a comparison of our technique with $\ell_1$ norm based methods.
>
> > The second contribution is to provide a simple algorithm for solving this objective. Past works have mostly relied on SGD/Adam to solve the FLOPs regularized objective. However, these algorithms are extremely slow to converge to sparse solutions and often require post processing steps. In our work, we overcome this problem by using proximal/projected techniques that are proven to converge to sparse solutions [1].
>
> $\textbf{Experiments on diverse models}$: We note that the experiments we presented in the paper already involve diverse models (BERT, MobileNet, EfficientNet, ResNet) and diverse layers (Transformer, Linear layers in BERT, Convolutions in MobileNet, EfficientNet). We also experimented with various compression building blocks including low-rank factorization, channel/neuron pruning, quantization, as shown in figures 1,3,4 and 5.
>
> $\textbf{Transfer learning and compression}$: We perform transfer learning and compression at the same time in our experiments, and add the FLOPs regularizer term to the cross-entropy loss during transfer learning.
>
>
> [1] Boyd, S.P. and Vandenberghe, L., 2004. Convex optimization. Cambridge university press.

---

> > ### Comment · Reviewer_qwYJ · 2023-11-23
> >
> > I appreciated the author's response to my comments.
> >
> > such solution is reasonable. I have increased the "Soundness" to 3. However, similar experiments or similar idea are well-established in previous studies, I still believe that the paper's novelty is not sufficient.

---

### Official Review · Reviewer_Hf33 · 2023-11-01

**Soundness:** 3 good
**Presentation:** 2 fair
**Contribution:** 3 good
**Rating:** 6
**Confidence:** 4

**Summary:**

In this paper, the authors propose an end-to-end framework for model compression. Specifically, the FLOPs is regarded as the target of optimization via l1/l2 latency surrogate. The experiments on MobileNetV3 and BERT show that the proposed method can achieve a cheaper architecture with almost the same amount of time as a single model training run.

**Strengths:**

1. This paper propose an end-to-end model compression method via optimizing the FLOPs.

2. The l1/l2 regularized surrogate is employed to optimize for the latency of the compact neural networks.

3. Experimental results on language and vision tasks demonstrate that the proposed method can find a more compact architecture.

**Weaknesses:**

1. The authors claim that the proposed method can be used with pruning, low-rank factorization, quantization etc. However, the quantization is not verified on the experiments.

2. The performance on different devices can be various, so I wonder is the proposed method still effective on other device?

**Questions:**

See weaknesses.

---

> ### Author Response · Authors · 2023-11-18
> **Response to the reviewer**
>
> We thank the reviewer for the valuable feedback.
>
> $\textbf{Quantization}$: We would like to note that we do have quantization results in the appendix (please see Figure 5 in the appendix). Due to space constraints, we didn’t place these experiments in the main paper.
>
> $\textbf{On device performance}$: In Appendix B.1 we describe how to take the device into account for model compression using our framework.  We develop a L1/L2 latency surrogate which takes the actual on-device latency into account. This helps directly optimize for on-device latency. In figure 6, the left plot presents the latency vs accuracy curves obtained using this technique for mobilenetv3 compression. The results show that working with latencies indeed helps us find models with better latencies than FLOPs regulrizer.

---

### Official Review · Reviewer_mcQZ · 2023-11-04

**Soundness:** 3 good
**Presentation:** 3 good
**Contribution:** 2 fair
**Rating:** 5
**Confidence:** 3

**Summary:**

This paper presents an end-to-end compression technique to compress deep models using a $l_1/l_2$ regularizer. The algorithm is versatile and fast and can be applied to different compression techniques, including pruning, low-rand factorization, and quantization. The authors build extensive experiments on various tasks, including BERT compression on GLUE fine-tuning tasks and MobileNetV3 compression on ImageNet-1K.

**Strengths:**

- The writing is clear and easy to understand.
- The proposed technique can be used with popular compression methods such as pruning, low-rank factorization, and quantization.
- The authors built experiments on various domains, including the pre-training and transfer learning tasks on CV and NLP benchmarks, like BERT compression on GLUE fine-tuning tasks and MobileNetV3 compression on ImageNet-1K.

**Weaknesses:**

- The main concern is the limited novelty of the work. The idea of making the FLOPs constraint differentiable is commonly used in many pruning works, such as using auxiliary masks. The positivity constraints are crucial for pruning models. However, as mentioned in the work, similar ideas have been studied in existing works.
- It is recommended to evaluate the performance of the proposed method multiple times, as it is claimed to be more stable than other methods.
- Direct comparison with advanced compression methods is also encouraged, and it would be better to show the training cost comparison since the proposed method is claimed to be fast.

**Questions:**

Please refer to the weakness.

---

> ### Author Response · Authors · 2023-11-18
> **Response to reviewer mcQZ**
>
> We thank the reviewer for the valuable feedback. Below we address some of the key concerns raised.
>
> $\textbf{Limited Novelty}$:  We would like to emphasize that introducing masks in the objective is not our contribution. Our main contribution is to address the challenges that arise in optimization of the resulting objective. Below we highlight our key contributions
>
> Our first contribution is to design a novel $\frac{\ell_1}{\ell_2}$ regularizer for optimizing the FLOP regularized objective. This helps us effectively compress models with batchnorm and layernorms. This is in contrast to prior works which use $\ell_1$ based FLOPs regularizer and as a result fail to work well in the presence of layer/batchnorms. Please see  Figure 2 in our paper for a comparison of our technique with $\ell_1$ norm based methods. We also compare against such techniques on a larger scale in figure 3(c), and empirically demonstrate our technique's better performance.
>
> The second contribution is to provide a simple algorithm for solving this objective. Past works have mostly relied on SGD/Adam to solve the FLOPs regularized objective. However, these algorithms are extremely slow to converge to sparse solutions and often require post processing steps, which include extra hyper-parameters for thresholding the sparse values.This limits their ease-of-use for practical deployment. In our work, we overcome this problem by using proximal/projected techniques that are proven to converge to sparse solutions [1].
>
> $\textbf{Multiple evaluations of our method}$: We would like to note that the experiments we performed on BERT, MobilenetV3, EfficientNet are large scale and it is computationally expensive to repeat these experiments multiple times. That being said, we note that we tried our technique on diverse tasks (please refer to experiments section in the paper) and noticed stable performance/loss curves across all the tasks.
>
> [1] Boyd, S.P. and Vandenberghe, L., 2004. Convex optimization. Cambridge university press.

---

> > ### Author Response · Authors · 2023-11-19
> > **Response to reviewer mcQZ (Part 2)**
> >
> > $\textbf{Comparison with advanced compression methods}$: We compare with state of the art compression methods on language and vision tasks, including recent works for compressing BERT, and compressing ResNet using pruning and NAS techniques. We also demonstrate that our method is much faster than state of the art NAS methods on mobileNet compression. In particular, we would like to highlight our results in figure 1, which compares our MobileNet pruning method against against TuNAS, an industry standard NAS algorithm which is widely deployed. We emphasize that MobileNetv3 and EfficientNet are highly optimized families of models, and our method can further compress them. We also compare against multiple recent state-of-the-art pruning and NAS works for ResNets in figure 3(a), and report much better performance. Finally, we also compare against recent state-of-the-art compression methods for BERT including [1] fine-tuning in figure 1, and report a better accuracy-FLOPs trade-off.
> >
> > [1] - Woosuk Kwon, Sehoon Kim, Michael W Mahoney, Joseph Hassoun, Kurt Keutzer, and Amir Gholami.
> > A fast post-training pruning framework for transformers. arXiv preprint arXiv:2204.09656, 2022.

---

> > > ### Comment · Reviewer_mcQZ · 2023-11-23
> > >
> > > Thank the authors' feedback. The authors address some of my concerns, but the novelty is limited. Thus, I keep my initial score.

---

> ### Author Response · Authors · 2023-11-23
> **Novelty of our work**
>
> We thank the reviewer for their response. We would like to follow up on the novelty aspect. In this work, we are addressing a practically relevant optimization aspect that is often neglected by the model compression/NAS community. We believe this is very important for making these algorithms more applicable in practice. If the reviewer believes these ideas have been explored in prior works, we would be very grateful if the reviewer can point us to some papers that have systematically tackled these issues.

---

### Meta-Review · Area_Chair_oD6g · 2023-12-08

**Metareview:**

This paper presents an end-to-end network compression technique that uses $l_1/l_2$ regularizer. Although the reviewers have recognized the importance of the problem and experimental validation, they have raised concerns regarding the novelty of the proposed approach. More specifically, Yang et al. (2019) and Diao et al. (2023) have explored $l_1/l_2$ regularization for network compression where instead of FLOPs, network sparsity is directly optimized. The rebuttal pointed out the differences (i.e., introducing FLOPs & improving optimizer in this work), however, the reviewers have found the improvements rather incremental (at least in the current presentation). Given these concerns, the submission is not ready for publication at ICLR. We hope that the feedback can help the authors improve this work for future conferences.

**Justification For Why Not Higher Score:**

The reviewers raised concerns regarding the novelty. The rebuttal couldn't address the concerns.

**Justification For Why Not Lower Score:**

N/A

---

### Decision · Program_Chairs · 2024-01-16

Reject